# Skeletal development in the sea urchin relies upon protein families that contain intrinsic disorder, aggregation-prone, and conserved globular interactive domains

Martin Pendola , Gaurav Jain, John Spencer Evans*

Laboratory for Chemical Physics, Center for Skeletal and Craniofacial Biology, New York University, New York, New York, United States of America

* jse1@nyu.edu

## Abstract

The formation of the sea urchin spicule skeleton requires the participation of hydrogel-forming protein families that regulate mineral nucleation and nanoparticle assembly processes that give rise to the spicule. However, the structure and molecular behavior of these proteins is not well established, and thus our ability to understand this process is hampered. We embarked on a study of sea urchin spicule proteins using a combination of biophysical and bioinformatics techniques. Our biophysical findings indicate that recombinant variants of the two most studied spicule matrix proteins, SpSM50 and SpSM30B/C (*S. purpuratus*) have a conformational landscape that include a C-terminal random coil/intrinsically disordered MAPQG sequence coupled to a conserved, folded N-terminal C-type lectin-like (CTLL) domain, with SpSM50 > SpSM30B/C with regard to intrinsic disorder. Both proteins possess solvent-accessible unfolded MAQPG sequence regions where Asn, Gln, and Arg residues may be accessible for protein hydrogel interactions with water molecules. Our bioinformatics study included seven other spicule matrix proteins where we note similarities between these proteins and rare, unusual proteins that possess folded and unfolded traits. Moreover, spicule matrix proteins possess three types of sequences: intrinsically disordered, amyloid-like, and folded protein-protein interactive. Collectively these reactive domains would be capable of driving protein assembly and hydrogel formation. Interestingly, three types of global conformations are predicted for the nine member protein set, wherein we note variations in the arrangement of intrinsically disordered and interactive globular domains. These variations may reflect species-specific requirements for spiculogenesis. We conclude that the molecular landscape of spicule matrix protein families enables them to function as hydrogelators, nucleators, and assemblers of mineral nanoparticles.

**Data Availability Statement:** All relevant data are within the paper.

**Funding:** The sole funder for this work is the U.S. Army Research Office. The funder had no role in

study design, data collection and analysis, decision to publish, or preparation of the manuscript. Authors Pendola, Jain, and Evans received partial salary from this funder.

**Competing interests:** The authors have declared that no competing interests exist.

**Abbreviations:** ACC, amorphous calcium carbonate; CDF, cumulative distribution function; CH, charge hydropathy; HSM30, spicule matrix protein 30, *Hemicentrotus pulcherrimus*; HSM41, spicule matrix protein 41, *Hemicentrotus pulcherrimus*; LSM34, spicule matrix protein 34, *Lytechinus pictus*; PM27, spicule matrix protein 27, *Heliocidaris erythrogramma*; PNC, pre-nucleation cluster; rSpSM30B/C-G, recombinant *Strongylocentrotus purpuratus* spicule matrix glycoprotein SpSM30B/C, B, C isoform hybrid; r-SpSM50, recombinant *Strongylocentrotus purpuratus* spicule matrix protein SpSM50; SpSM29, spicule matrix 29, *Strongylocentrotus purpuratus*; SpSM32, spicule matrix 32, *Strongylocentrotus purpuratus*; SpSM37, spicule matrix 37, *Strongylocentrotus purpuratus*.

# Introduction

The formation of endo- and exoskeletons requires the participation of protein families that enable the construction of biomaterials that can withstand stress and provide support, protection, and survival [1–3]. Among these are the sea urchin skeletal elements, e.g., the protective embryonic spicules and adult spines where extracellular matrix (ECM) proteins combine to form an environment for inorganic nanoparticle nucleation (calcium carbonates), nanoparticle assembly, and the creation of a mineralized fracture resistant inorganic-organic composite [1–3]. In the sea urchin embryo, the spicule matrix (SM) proteome plays an important role in creating a hydrogel-based meshwork within the ECM that limits ion diffusion, creates ultra-small volume compartmentalization for nanoparticle formation, and assembles the nanoparticles into a mesocrystal [4–7]. Several SM proteins have been identified as regulators of the spicule mineralization process [4–17] and some of these protein sequences in peptidomimetic form have been shown to possess very interesting conformational properties, such as the presence of intrinsic disorder or unfolded structure, repetitive beta turn, glycine loop, and extended twist structural repeats [18–20]. In studies of two *Strongylocentrotus purpuratus* spicule matrix proteins, SpSM30B/C [13] and SpSM50 [14] it was speculated that the molecular features which promote protein-protein assembly, matrix formation, water binding and release, and nanoparticle assembly [13–17] include disordered [21–29] and amyloid-like [30–32] sequences. It is likely that conserved SM domains, such as the C-type lectin-like (CTLL) carbohydrate binding motif [7–12], are also contributors to these processes. However, the structural features of SM protein families have not been fully identified, either in *S. purpuratus* [4,7–12] or in other sea urchin species, and important functionalities such as hydrogel-water binding [17] have not been fully explained. Thus, further investigation is required if we are to understand protein matrix formation and hydrogel regulation of mineralization during spicule development.

In this article, we address the structural features that exist amongst SM proteins, and, identify the basis for spicule matrix protein hydrogel-water binding and release. To achieve this, we first performed experiments on a well-characterized set of hydrogelator SM proteins from the well-known *Strongylocentrotus purpuratus* proteome (SpSM50, pI = 10.7, 428 AA, 44541 Da; SpSM30B/C, pI = 5.73; 270 AA, MW = 33287.4 Da)[7–17]. Using recombinant, tag-free variants [13,14] of both proteins (rSpSM50, rSpSM30B/C-G) we determined that each protein is intrinsically disordered and possesses residual secondary structures within the assembled hydrogel state. Additionally, in hydrogel form both rSpSM50 and rSpSM30B/C-G proteins exhibit solvent-accessible Asn, Gln, and Arg sidechain residues and these residues are likely candidates for observed water exchange and subsequent mineral precursor hydration/dehydration processes reported in earlier studies [15–17]. Subsequently, we complemented our experimental studies with bioinformatics investigations of a subset of nine published SM sequences originating from four different sea urchin species (*Lytechinus pictus*, *Hemicentrotus pulcherrimus*, *Strongylocentrotus purpuratus*, *Heliocidaris erythrogramma*)[33–36]. Here, the idea was to extend our SpSM50 and SpSM30B/C experimental investigations, by using predictive bioinformatics [25–27;30–32;37,38] to determine the presence of hydrogelator-related structural traits, such as intrinsic disorder, aggregation propensity, and interactive conserved folded sequence regions [13,14] within other spicule matrix proteins. Together, these approaches revealed that spicule matrix protein families have a common molecular landscape that features an open global conformation consisting of intrinsically disordered, amyloid-like cross-beta strand, and folded protein-protein interactive domains. It is likely that these molecular features not only drive protein hydrogel formation, but also spiculogenesis and biomineralization schemes within the skeletal development of sea urchins in general.

## Experimental

### Sample preparation

The expression, preparation and purification of recombinant tag-free insect cell expressed SpSM30B/C-G glycoprotein (rSpSM30B/C-G) and bacteria-expressed SpSM50 (rSpSM50) were performed as described previously [13,14]. For subsequent experimentation, both protein samples were created by exchanging and concentrating appropriate volumes of stock solution into unbuffered deionized distilled water (UDDW) or other appropriate buffers using Amicon Ultra 0.5 3 kDa MWCO concentration filters [13,14]. For subsequent experiments protein concentrations were determined using rapid gold BCA protein assay kit (ThermoFisher Scientific, USA).

### Circular dichroism spectrometry

CD spectra (190–260 nm) of 3 μM rSpSM50 and 7.5 μM rSpSM30B/C-G in 100 μM HEPES pH 8.0 were collected at 25 ˚C on the AVIV Stopped Flow 202SF CD Spectropolarimeter [39]. Due to the high aggregation propensity of rSpSM50, this protein was examined at a lower protein concentration (~ 2.5x) compared to rSpSM30B/C-G. A total of eight scans per sample were collected in a cuvette with 0.1 cm path length, using 1 nm bandwidth, 1 nm wavelength step and 0.5 s averaging time. The instrument was previously calibrated with d-10-camphorsulfonic acid [39]. The recorded spectra were averaged and the appropriate background spectra (HEPES buffer) subtracted. Spectra were smoothened using the binomial algorithm included in the AVIV CD software. Ellipticity is reported as mean residue ellipticity (deg cm$^2$ dmol$^{-1}$)[39].

### NMR spectroscopy

We performed $^1$H NMR PFG experiments on 22 μM rSpSM30B/C-G and rSpSM50 in 150 μL of 30 nm filtered Fisher Ultrapure water (Fisher Scientific, USA) containing 10% v/v 99.9% $D_2O$ (Cambridge Scientific Labs, USA) and 100 μM HEPES, pH 7.5 [17]. The presence of HEPES buffer induces hydrogelation of both proteins [13,14]. The purpose of these experiments was to determine if there exist any labile or mobile backbone and sidechain regions within protein molecules that comprise the hydrogel particles. $^1$H NMR experiments were conducted at 25 ˚C on a Bruker AVANCE-800 NMR Spectrometer using a 3 mm cryo-probehead. 2-D TOCSY (mixing time = 40 msec) and NOESY (mixing time = 50 msec) experiments [40,41] were performed on all samples using the following parameters: 16 scans per experiment; relaxation delay = 1.5 sec; WATERGATE gradient solvent suppression. All NMR data was processed, analyzed, and plotted using TopSpin Software (Bruker BioSpin, USA) and $^1$H NMR chemical shifts are reported from internal d$_4$-TSP (deuterated trimethylsilapentanesulfonic acid)[17,40,41].

### Bioinformatics

Intrinsic disorder [DISOPRED [25], IUP [26], GLOBPLOT 2.3 [27]) and short length amyloid-like cross-beta strand sequence (FOLD_AMYLOID [30], AGGRESCAN [31], ZIPPER_DB [32]) prediction algorithm cohorts were utilized using default parameters to comparatively map out intrinsic disorder and aggregation-prone sequences, respectively, for nine spicule matrix protein sequences (Table 1). Putative signal peptide regions were identified using ExPASy Signal P software [42] and these signal regions were deleted from each DNA-derived sequence prior to the analyses described above. The charge–hydrophobicity plots (CH-plots) [43] and the cumulative distribution function (CDF) analyses [44] were used for

**Table 1. Spicule matrix protein sequences.**

| Spicule Matrix Protein | Sea Urchin Species | Accession Number |
|---|---|---|
| SpSM50 | *Strongylocentrotus purpuratus* | P11994, SM50_STRPU |
| SpSM37 | *Strongylocentrotus purpuratus* | Uniprot O76450, GenBank AAC33762.1 |
| SpSM32 | *Strongylocentrotus purpuratus* | Uniprot Q8MUL1, GenBank AAM70486.1 |
| SpSM30B/C | *Strongylocentrotus purpuratus* | P28163, SM30_STRPU |
| SpSM29 | *Strongylocentrotus purpuratus* | Uniprot Q8MUL0, GenBank AAM70487.1 |
| LSM34 | *Lytechinus pictus* | Uniprot Q05904, GenBank CAA42179.1 |
| HSM30 | *Hemicentrotus pulcherrimus* | Uniprot Q25116 |
| HSM41 | *Hemicentrotus pulcherrimus* | Uniprot Q26264, GenBank AAB24285 |
| PM27 | *Heliocidaris erythrogramma* | Uniprot Q95W96 |

binary prediction of protein stability based of its amino acid sequence; these values were calculated using PONDR® online service (http://www.pondr.com/) to create the CH-CDF analysis [45]. To determine a hypothetical global structure of each SM sequence, we utilized the DISO-clust—IntFOLD4 integrated protein structure and function prediction server (University of Reading, UK, using default parameters [37,38] which provides tertiary structure prediction/3D modeling of protein sequences that contain folded and unfolded sequence elements. Molecules were visualized using PyMol (Schrodinger, Pasadena, CA, USA).

## Results

rSpSM30B/C-G and rSpSM50 possess different contents of intrinsic disorder and residual secondary structure in the gel state. Bioinformatics have indicated that SpSM30B/C and SpSM50 proteins contain a folded CTLL domain at the N-terminus and an unfolded, MAQPG repetitive sequence domain at the C-terminus [13,14]. To explore this, we employed circular dichroism spectrometry and examined the recombinant versions of both proteins under conditions known to promote hydrogel particle formation (10 mM HEPES, pH 8.0)(Fig 1) [13,14]. We find that in the suspended gel state rSpSM50 presents with a single (-) $\pi$– $\pi^*$ transition minima band centered near 222 nm, and rSpSM30B/C-G exhibits with a single (-) $\pi$– $\pi^*$ transition minima band centered near 216 nm, i.e., a 6 nm blue shift from rSpSM50. In either case these transition minima bands are consistent with the presence of intrinsically disordered conformations along with residual secondary structures such as alpha-helix and beta-strand [18–20;39]. From this data we conclude that both proteins in the gel state possess residual secondary structure and disordered or unfolded structural content. If we presume that intrinsic disorder content is a contributing driving force in self-association [22–25], then this in part explains why both sea urchin spicule matrix proteins are strong aggregators that can form disordered gels [13,14,17].

SpSM protein hydrogels possess accessible Asn, Gln, Arg residues. It has been established that rSpSM30B/C-G and rSpSM50 hydrogels can rapidly exchange water [17]. This ability may be crucial for hydration-based processes in nucleation such as stabilization/destabilization of amorphous calcium carbonate [13–15;17]. But to achieve this, the hydrogel particles must possess regions where mobile protein sidechains can access and bind/release solvent molecules, e.g., on the exterior surface or within the porous interior of the hydrogel. Previously, NMR spectroscopy of mollusk shell-associated biomineralization protein hydrogels revealed that not all of the amino acid residues are involved in intermolecular contacts; rather, some of these are mobile and therefore accessible, either internally or externally, for additional interactions such as solvent or solute binding [40,41]. This mobility/accessibility phenomenon may also exist in SpSM hydrogel particle systems [17] but has yet to be investigated.

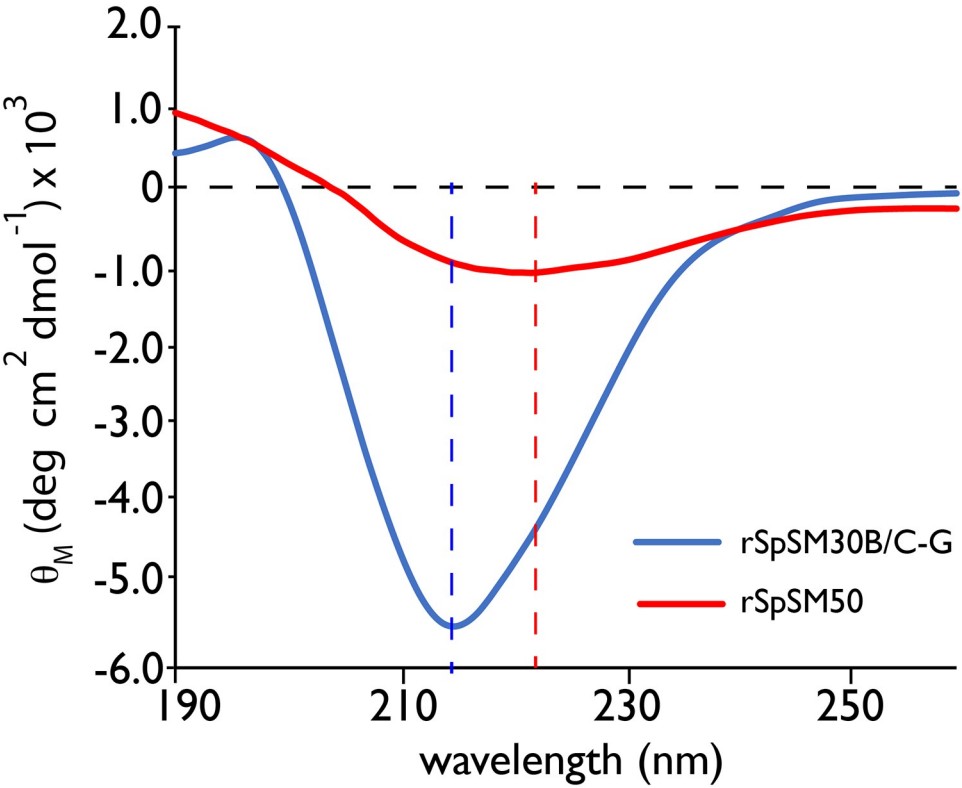

**Fig 1. Far-UV circular dichroism spectra of rSpSM30B/C-G (7.5 µM) and rSpSM50 (3 µM) proteins in 100 µM HEPES, pH 8.0.** Dashed line extrapolates the ellipticity minima for each protein.

To determine if rSpSM50 and rSpSM30B/C-G protein molecules within hydrogel particles possess backbone or sidechain mobility, we performed $^1$H NMR experiments on both samples under conditions which promote hydrogelation (pH 7.5 in 100 µM HEPES)(Fig 2) and examined the exchangeable sidechain NH proton frequency region (6.5–8.0 ppm). We chose $^1$H NMR as a simple, low cost approach to understanding sidechain molecular mobility and accessibility [40] in these protein molecules and to plan for future $^{13}$C/$^{15}$N labeled multidimensional NMR studies [41] of these proteins. Note that due to the use of HEPES buffer, $^1$H NMR chemical shift overlap generated by HEPES $^1$H resonances prevented us from performing analyses of

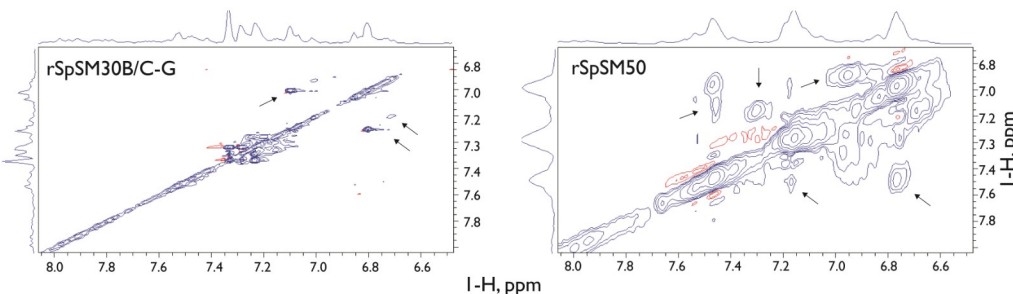

**Fig 2. Homonuclear 800 MHz $^1$H TOCSY spectra (exchangeable sidechain amide chemical shift region) of 22µM rSpSM30B/C-G and rSpSM50 hydrogel particle samples, 100 µM HEPES, pH 7.5.** Diagonal and off-diagonal regions for sidechain and backbone NH Arg, Asn, and Gln resonances are shown, along with corresponding 1-D spectra.

the aliphatic proton sidechain region (i.e., 0–5 ppm) and thus our focus is limited to the NH sidechain frequency region. As expected, intermolecular contacts between protein molecules within large rSpSM50 and rSpSM30B/C-G hydrogel assemblies would be expected to attenuate the majority of NH backbone and sidechain resonances in the TOCSY spectra for both samples, since aggregation-induced changes in protein backbone dynamics leads to intermediate time scale broadening and other relaxation effects [40,41]. Surprisingly, a closer look reveals that not all of the NMR NH sidechain resonances are attenuated in either the rSpSM50 or rSpSM30B/C-G samples (Fig 2). The absence of signal attenuation indicates that these sidechain resonances are not involved in intermolecular contacts within the protein hydrogels or limited in molecular motion. Thus, there are some amino acids that reside in rSpSM50 and rSpSM30B/C-G protein sequence regions where mobility is permissible and thus may reside in solvent-accessible interior or exterior regions of the hydrogel particles [40,41].

Although NMR signal attenuation and $^1$H NMR chemical shift overlap prevents us from obtaining sequence-specific spectral assignments at this time, we can leverage the unique repetitive sequence features of both proteins to identify the types of amino acids within each rSpSM protein that give rise to these resonances. In both the rSpSM30B/C-G and rSpSM50 TOCSY spectra we note scalar crosspeaks in the $^1$H NMR frequency range of 6.80–7.40 ppm, which are consistent with hydrogen-bonding donor/acceptor Arg δ-NH guanidine sidechain proton resonances (Fig 2). Additionally, we also note the presence of TOCSY crosspeaks in the $^1$H NMR frequency range of 7.1–7.4, which are consistent with hydrogen-bonding donor/acceptor Asn δ-NH and Gln ε-NH amide sidechain resonances (Fig 2). The content of these three amino acids is significant within the unfolded MAQPG C-terminal sequence region of each protein (rSpSM30B/C-G = Asn, 55%, Gln, 47%, Arg = 58%; rSpSM50 = Asn, 94%, Gln = 75%, Arg = 89%)(Fig 3)[13,14]. Thus, our TOCSY experiments suggest that the non-attenuated Asn, Gln, and Arg NH resonances most likely arise from the unfolded C-terminal MAQPG sequence regions of rSpSM30B/C-G and rSpSM50 molecules that reside throughout the hydrogel matrices. Since the unfolded MAQPG regions would be expected to be motionally unrestricted (i.e., greater degrees of freedom) and solvent-accessible [13,14,17], then the Asn, Gln, and Arg residues in these regions would also be expected to exhibit molecular mobility and solvent-accessibility as well. Thus, we hypothesize that the MAQPG domain and its associated Asn, Gln, Arg residues represents a putative site for rapid water exchange phenomena that we observed for both spicule matrix protein hydrogels in bulk solution [17]. Note that at this time we are unable to determine if other MAQPG—associated hydrogen-bonding donor-acceptor amino acids, such as His, Thr, Ser, and Tyr [13,14], are also solvent-accessible within rSpSM50 and rSpSM30B/C-G hydrogels and participate in water exchange as well.

Molecular landscape: SM sequences are defined by disorder content, aggregation propensity, and folded/unfolded structure. Although studies have shown that both SpSM30B/C and SpSM50 possess an intrinsically disordered MAQPG domain and aggregation-prone sequences [13,14] little is known regarding the extent of intrinsic disorder content or aggregation propensity within other spicule mineral-associated proteins that are expressed during embryonic development in *S. purpuratus* or in other sea urchin species. To explore this further, we initiated a bioinformatics study to determine the frequency and location of intrinsically disordered (DISOPRED [25], IUP [26], GLOBPLOT 2.3 [27]) and short length amyloid-like cross-beta strand aggregation propensity sequences (FOLD_AMYLOID [30], AGGRESCAN [31], ZIPPER_DB [32]) within nine spicule matrix protein sequences obtained from four different sea urchin species (*Lytechinus pictus*, *Hemicentrotus pulcherrimus*, *Strongylocentrotus purpuratus*, *Heliocidaris erythrogramma*)(Table 1, Fig 4).

We will first consider the issue of unfolded or disordered states (Fig 4). We found that all nine spicule matrix protein sequences contain varying percentages of intrinsic disorder, with

## SpSM30B/C-G

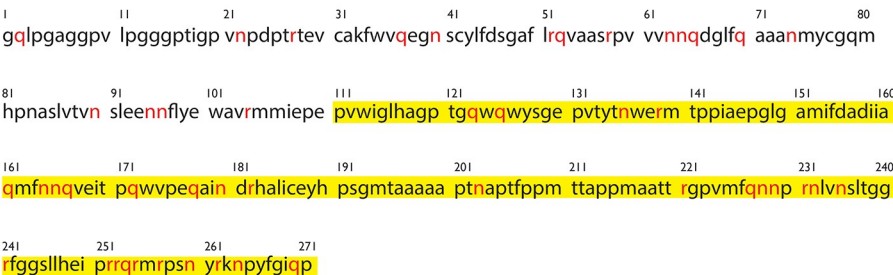

## SpSM50

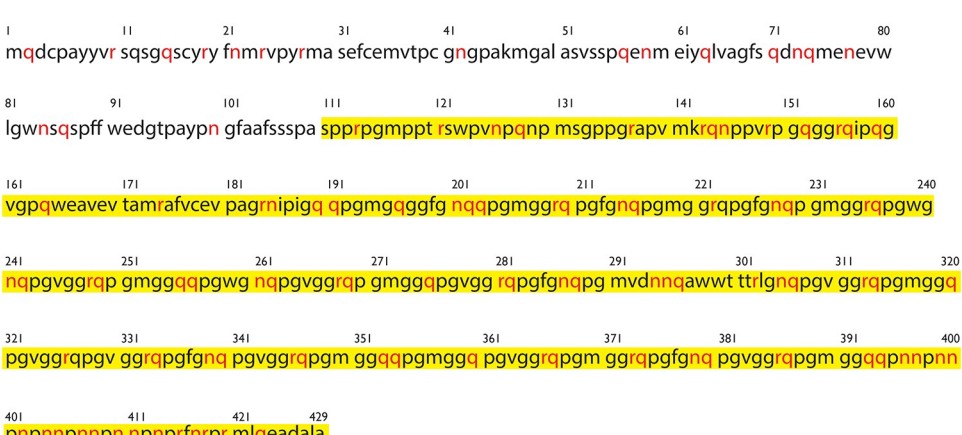

**Fig 3. Primary sequences of SpSM30B/C-G and SpSM50.** Arg, Gln, and Asn residues are presented in red. MAQPG domains are highlighted in yellow. Note high concentration of Arg, Asn, Gln within disordered MAQPG regions.

SpSM50 and HSM30 possess the largest (75%) and smallest (22%) sequence percentage of intrinsic disorder, respectively, and the average percentage of intrinsic disorder content being 40% for all spicule matrix proteins in this study. Subsequently we calculated the CH-CDF (charge hydropathy—cumulative distribution function) scores for all nine spicule matrix protein sequences (Fig 5)[44,45]. CH-CDF plots provide comparisons of structure-disorder tendencies within proteomes and is more sensitive to disorder than a traditional CH plot. Here, we observe a narrow distribution of CH scores (i.e., > 0.39, < 0.47) for the nine proteins, indicating that charge-hydropathy values are similar amongst these sequences. In contrast, what distinguishes the spicule matrix sequences from one another are their CDF scores, which exhibit a broader distribution (i.e., > 0.4, < 0.9) compared to the CH scores. We see that SpSM50 and SpSM30B/C have the lowest and highest CDF scores, respectively, indicating that SpSM50 is more unfolded than SpSM30B/C. But what is remarkable about Fig 5 is when we integrate the CH and CDF scores together, we note that all spicule matrix sequences fall into the Quadrant 1 region, which represents rare or unusual proteins whose CDF scores correspond to folded proteins but the CH scores are typical of disordered sequences [44,45]. At this time there is very little data available for Quadrant 1 type proteins [44,45], but it should not be surprising that spicule matrix proteins fall into this rare category given their unique repetitive

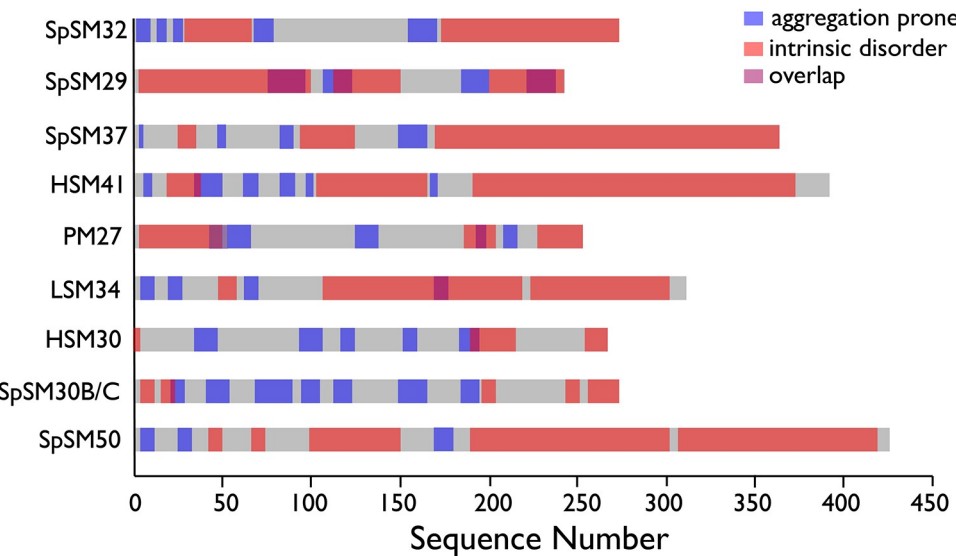

**Fig 4. Predicted regions of intrinsic disorder (GLOBPLOT 2.3, DISOPRED, IUP) and aggregation-prone amyloid-like (AGGRESCAN, FOLD_AMYLOID, ZIPPER_DB).** Shaded areas (red = intrinsic disorder; blue = amyloid-like cross-beta strand) denote sequence regions predicted as positive by each cohort of algorithms. Grey area denotes regions that do not score as positive for either intrinsic disorder or amyloid-like sequences. Purple color denotes sequence region overlap between aggregation-prone and intrinsic disorder.

disordered MAQPG sequences, presence of modified globular domains [4,7–12], and in the case of SpSM50 and SpSm30B/C, their hydrogelation propensities.

The significance of intrinsic disorder within spicule matrix protein sequences is that these regions are energetically unstable due to the absence of stabilizing elements such as intrastrand backbone hydrogen bonding found in alpha-helical and beta-strand sequences [22–25]. In some cases, it has been documented that some unstable disordered domains can be triggered to fold when they bind to targets or are influenced by environmental factors [23–25]. In either instance, sequence reactivity would be important for promoting protein—protein interactions that lead to hydrogelation and the formation of protein hydrogel particles [13–17].

We now examine aggregation propensity. All nine spicule matrix protein sequences possess more than one aggregation-prone amyloid-like cross-beta strand region, with SpSM30B/C and SpSM29 possess the highest (7) and lowest (4) number of regions, respectively (Fig 4), with the average number of amyloid-like domains being five for all proteins in this study. The significance is that these aggregation-prone sequences have been shown to be important for initiating molecular assembly [30–32]. The widespread occurrence of these short motifs within the nine tested SM proteins strongly suggests that amyloid-like aggregation motifs may play an important role in spicule matrix assembly and hydrogel formation [13–17].

However, Figs 4 and 5 are two-dimensional and tell only part of the story. The overall molecular features which enable spicule matrix proteins to self-assemble and form protein hydrogel particles are three-dimensional in nature and can be better understood using algorithms such as the DISOclust/IntFOLD4 predicted 3D modeling prediction program [37,38]. Here, the algorithm uses sequence homology modeling and intrinsic disorder prediction to generate a qualitative global conformation for proteins that possess both folded and unfolded regions (Fig 6). A comparison of the predicted global conformations of the nine spicule matrix proteins (best template model for the globular domain, confidence levels, P scores, and global model quality scores, see Table 2) in this study reveal some interesting trends for global

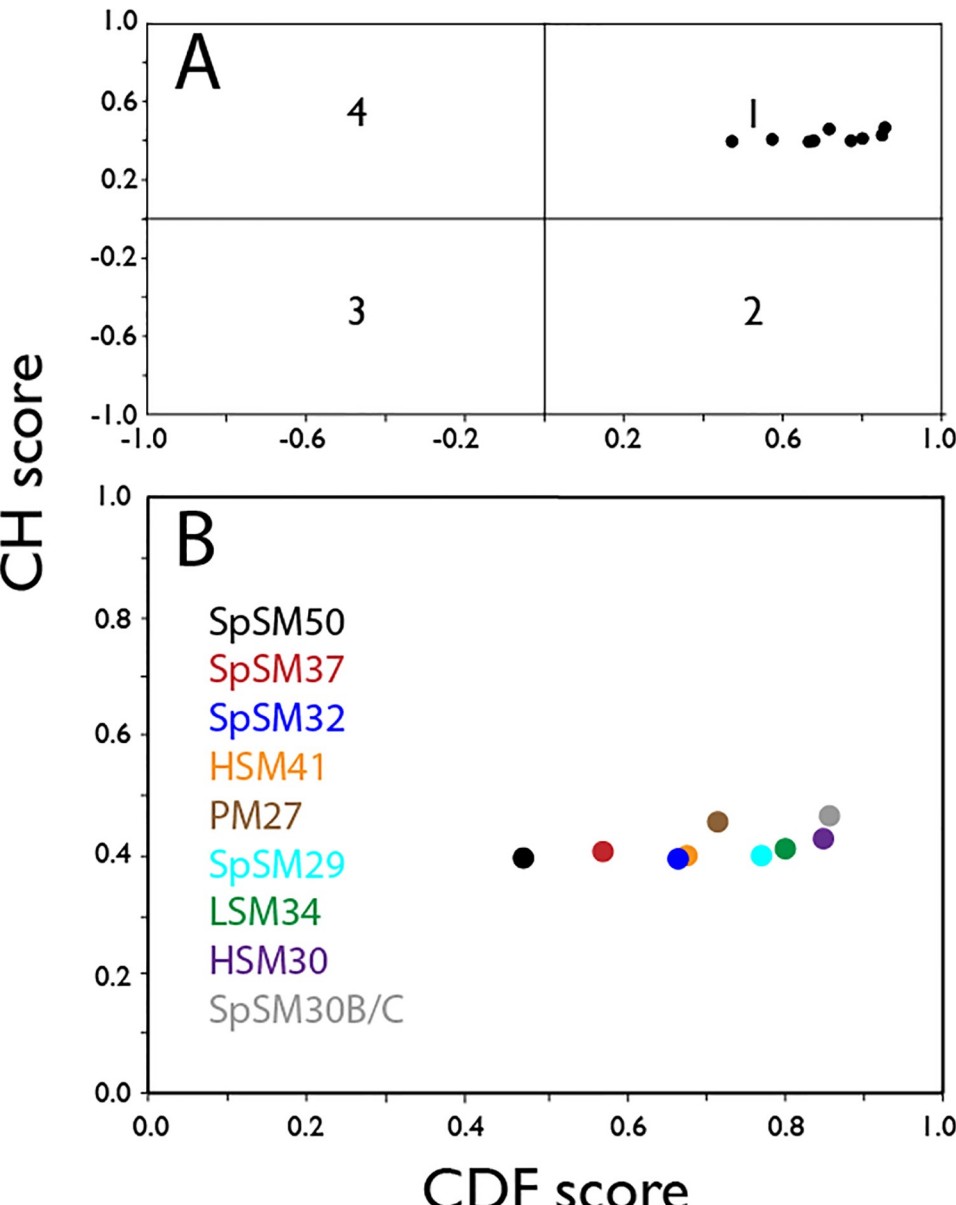

**Fig 5.** (A) Four quadrant (1–4) CH-CDF plot for spicule matrix protein sequences. (B) Enlargement of relevant Quadrant 1. The Y-coordinate in the CH-CDF plot corresponds to the distance from the obtained ordinate value to the correlation line separating the structured and unstructured conformational state of the protein on the CH (charge-hydrophobicity) plot. The X-coordinate on the CH-CDF plot corresponded to the distance from the obtained ordinate value to the correlation line separating the structured and unstructured conformational state of the protein in the CDF. There are 4 quadrants: Quadrant 1 (CH > 0, CDF > 0) representing rare proteins for which it is impossible to determine accurately the state, i.e., their CDF scores correspond to structured domains but CH scores correspond to unstructured proteins. Quadrant 2 (CH > 0, CDF < 0) represents unfolded proteins (U), Quadrant 3 (CH < 0, CDF < 0) represents the molten globule state (MG). Quadrant 4 (CH < 0, CDF > 0) represents structured or folded proteins (F)[44,45].

conformations. First, it is clear that open, unfolded conformations comprise a significant proportion of the global structure of each protein, which supports the findings obtained in Figs 4 and 5. Given that these proteins assemble to form matrices, an open unfolded global conformation would facilitate protein-protein interactions that are necessary for matrix formation.

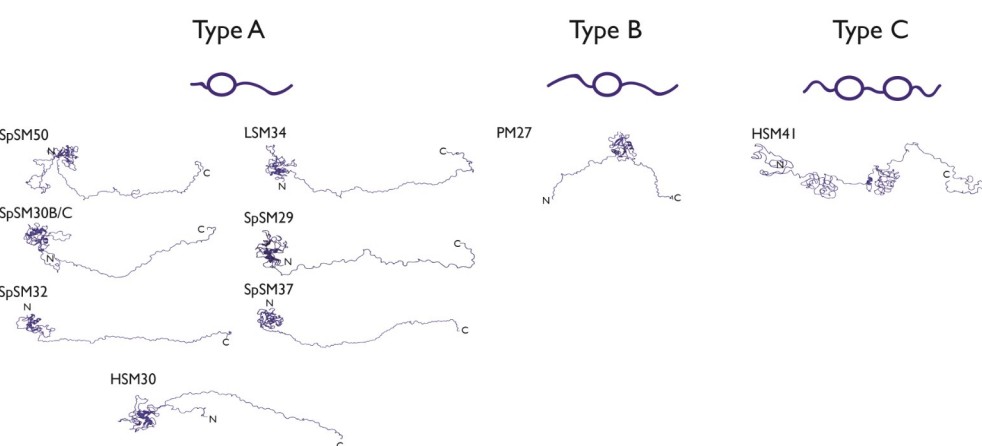

**Fig 6. Categories of spicule matrix protein backbone conformations predicted by DISOclust/Intfold 4.0 (ribbon representation, lowest energy conformer) for nine sea urchin spicule matrix proteins (Table 1).** Under each Type is a cartoon representation of global conformation (circle = folded conformation; squiggle line = disordered conformation). Best template model for the globular domain, confidence levels, P scores, and global model quality scores can be found in Table 2. N- and C-terminal ends are denoted.

However, of equal importance is the presence of a conserved, interactive CTLL domain (Table 2) in eight of the nine proteins, which we believe is also involved in matrix formation. Interestingly, the outlier to this trend is HSM41; instead of a single CTLL domain, this protein contains two globular, folded regions that are known protein-protein interaction domains: the fatty acid synthase α-subunit of *Saccharomyces cerevisiae*, and the PSCD-region of the cell wall protein pleuralin-1 from the biomineralizing silica diatom, *Cylindrotheca fusiformis* (Table 2). Like the CTLL domain, we believe that these two interactive folded domains represent putative sites within HSM41 for spicule matrix assembly. Obviously, the substitution of these two

**Table 2. DISOclust/INTFOLD4 fitted crystal structure template models homologous to conserved globular domains in sea urchin spicule matrix proteins.**

| Protein | Model Template (globular domain) | Confidence/P value | Global model quality score |
|---|---|---|---|
| SpSM50 | 3alsA | High/3.23 E-3 | 0.5112 |
| SpSM30B/C | 1qddA, 1jznA, 1eggB | High/4.53 E-3 | 0.5092 |
| SpSM37 | 3alsA | Medium/1.552 E-2 | 0.3794 |
| SpSM32 | 1wmyA, 1jzna | High/2.684 E-3 | 0.4205 |
| SpSM29 | 2ox9C | Cert/2.406 E-5 | 0.5333 |
| LSM34 | 1wmyA | High/5.13 E-3 | 0.4055 |
| HSM41 | 2nbiA, 2pff | High/8.79 E-3 | 0.3920 |
| HSM30 | 1qddA, 1jznA | High/6.215 E-3 | 0.4788 |
| PM27 | 1wmyA | Cert/4.182 E-5 | 0.5158 |

3alsA, 1wmyA = C-type lectin CEL-I, *Cucumaria echinata*

1jznA = Galactose-specific C-type lectin, *Crotalus atrox*

1qddA = Lithostathine, *Homo sapiens*

2h2r = CD23 lectin domain, *Homo sapiens*

2ox9C = Mouse scavenger receptor C-type Lectin carbohydrate-recognition domain, *Mus musculus*

1eggB = C-type carbohydrate recognition domain (CRD-4), macrophage mannose receptor, *Homo sapiens*

2pff = fatty acid synthase subunit alpha, *Saccharomyces cerevisiae.*

2nbiA = pscd-region of the cell wall protein pleuralin-1, *Cylindrotheca fusiformis*

globular domains for the *S. purpuratus* CTLL domain represents a species-specific adaptation of HSM41 for spicule matrix formation in *H. pulcherrimus*, which we will discuss later on.

Analyses of these predicted SM global structures reveal 3 conformational categories (arbitrarily denoted as Types A, B, C) that reflect two variables: 1) the position of disordered regions relative to conserved globular domains, and 2) the number of conserved globular domains. The Type A conformation is represented within the entire SpSM series as well as HSM30 and LSM34 (which is homologous to SpSM50)[40]. The Type A conformer consists of a short (~5–40 AA) disordered N-terminal segment, followed by a single conserved globular CTLL domain, and lastly by an open conformation C-terminal MAQPG domain. In contrast, Type B conformation, which is represented by PM27, has a longer N-terminal disordered sequence ($\geq$ 50 AA) that is coupled to a single conserved CTLL domain, which is then followed by an open conformation C-terminal MAQPG domain. In both classifications, the presence of extended, disordered MAQPG regions can act as interactive motifs for protein-protein binding [13,14] and sidechain—water interactions [17]. Lastly, Type C, represented by HSM41, is essentially a modified Type B configuration: two globular folded domains linked by an open conformation region with flanking N- and C-terminal regions existing in an open conformation. From these results, we conclude the following: 1) Given the hydrogelation capabilities of SpSM50 and SpSM30B/C [13–17], the gross similarities of the Type A, B, C conformations (i.e., interactive globular + reactive disordered) suggest that all investigated spicule matrix proteins are hydrogelators as well. 2) The noted differences in globular—disordered domain sequence locations and the length of disordered sequences (Figs 3 and 5) could affect a number of parameters with regard to spicule matrix protein hydrogels, such as pore size, gel density, nanoparticle formation and ordering, and intracrystalline nanoinclusion size and distribution, features which we have noted to be unique for SpSM50 and SpSM30B/C [13–17].

## Discussion

Recently, considerable emphasis has been placed on intrinsic disorder as a major factor in the formation of skeletal extracellular matrices, particularly those that support the biomineralization process [22,23,28]. Given that intrinsically disordered sequences lack internal stabilization (e.g., absence of intrastrand hydrogen bonding), they are thermodynamically unstable and hence highly reactive to other molecular species such as proteins or other substrates [21–28]. This would explain the presence of intrinsic disorder within extracellular matrix protein sequences that enable protein assembly. However, based upon our current study we believe that intrinsic disorder represents only part of the matrix story. The spicule matrix proteome consists of 3 major components: a) intrinsically disordered regions; b) folded protein-protein interactive motifs, and c) amyloid-like cross-beta strand aggregation-prone sequences (Figs 1, 4 and 5; Table 2). Similar results been reported in different mollusk shell biomineralization protein investigations [22;39–41]. We argue that there is a need for this structural heterogeneity in biomineralization proteins: 1) The creation of a hydrogel matrix that can bind and release water, thereby impacting the mineralization process [19] requires protein-protein recognition and assembly [13–17;39–41]. Amyloid-like and interactive globular domains jointly satisfy this requirement alongside intrinsically disordered domains. 2) There is also a requirement to generate a disordered gel or polymer induced liquid phase (PILP) matrix that acts as a liquid-liquid phase separator [46] for the formation and assembly of mineral phase precursors, such a pre-nucleation clusters [47,48], into amorphous calcium carbonate (ACC)[46–48]. In this instance, intrinsically disordered sequences are appropriate. We postulate that the number and location of intrinsically disordered, amyloid, and conserved domains within specific

spicule matrix proteins (Figs 4 and 6) reflects the number and nature of potential matrix molecular species that each protein is destined to interact with [16].

Our study of rSpSM50 and rSpSM30B/C-G highlights the structural similarities and differences that exist within the two major spicule matrix proteins expressed by the embryonic sea urchin *S. purpuratus*. From previous studies we learned that rSpSM50 is a stronger aggregator than rSpSM30B/C-G, forming larger dimension hydrogel particles in solution [13,14] and inducing a greater degree of mineral particle organization [15]. Obviously, factors such as molecular net charge (rSpSM50 = cationic; rSpSM30B/C-G = anionic) and the presence of glycosylation (rSpSM30B/C-G) could explain these differences in hydrogelation [13,14]. However, based upon our CD (Fig 1) and bioinformatics (Figs 4, 5 and 6) studies, we believe that differences in structural features also play a role in defining rSpSM50 and rSpSM30B/C-G aggregation and the organization of hydrogel particles. Specifically, rSpSM30B/C-G possesses a higher degree of residual secondary structure and lower degree of intrinsic disorder relative to rSpSM50 (Figs 4 and 5). Given that rSpSM50 > rSpSM30B/C-G in terms of aggregation and hydrogel particle size [13,14], we would conclude that intrinsic disorder content plays an important role in spicule matrix protein hydrogelation and explains in part why rSpSM50 exhibits higher aggregation propensity. In turn, the differences in protein hydrogelation could impact calcium carbonate nucleation, intracrystalline nanoporosity size and distribution, solute and water diffusion, and mineral particle assembly and organization [13–17].

Although structural differences exist between rSpSM50 and rSpSM30B/C-G (Fig 1), they essentially function on an equivalent level with regard to water exchange [17], and this is reflected in our NMR data, where both proteins possess detectable hydrogen-bonding donor/acceptor amino acids (e.g., Asn, Gln, Arg) (Figs 2 and 3) within the motionally unrestricted, solvent-accessible intrinsically disordered MAQPG regions (Figs 2 and 3). This implicates the MAQPG region and its associated Asn, Gln, Arg residues as one site for rapid water exchange [17] throughout the protein hydrogel network created by rSpSM50 and rSpSM30B/C-G [13–16]. As described in our past work, this type of exchange could affect the hydration or solubility of precursor calcium carbonate mineral phases during nucleation [13–15], which, in turn, could affect amorphous calcium carbonate stabilization and eventual transformation into calcite [46–48] is process would be highly relevant for spiculogenesis and biomineralization and we believe that other spicule matrix protein hydrogels (Fig 6, Table 1) would likewise possess similar features that can engage in water exchange processes.

We can use the insights obtained from the SpSM50 and SpSM30B/C data (Figs 1 and 2) along with bioinformatics predictions (Fig 6) to extend our understanding of structure and function within the known *S. purpuratus* proteome. In general, the spicule matrix proteome is similar to rare, unusual proteins whose traits mimic both folded and unfolded species (Fig 5) [44,45]. A closer examination (Fig 6) reveals how disordered and folded traits are utilized in these proteins. Here, we note that the predicted Type A conformation, featuring the interactive CTLL domain, represents the majority of known spicule matrix protein sequences originating from *S. purpuratus* (Fig 6)[7–12]. What distinguishes different Type A proteins from one another is the structure of the interactive CTLL domain itself (Table 2), which we note to be similar in SpSM50 and SpSM37 yet different in SpSM32, SpSM29, and SpSM30B/C. We interpret these findings as follows: 1) the gross conformational similarities within Type A signal functional similarities in these proteins vis a vis hydrogel formation, water exchange, and subsequent spicule mineralization in *S. purpuratus*. 2) The variations in CTLL domain structure (Table 2), intrinsic disorder and amyloid-like sequences (Fig 4) may be indicative of slightly different molecular recognition strategies used by different *S. purpuratus* proteins to identify and bind to each other during spiculogenesis, such as SpSM50 binding to SpSM30B/C [16].

But what of other sea urchin species? Once again, the rare, unusual traits of disorder, aggregation propensity, and folded structure manifest themselves in other sea urchin species. We find that LSM34 (*L. pictus*) and HSM30 (*H. pulcherrimus)* are conformationally similar to the *S. purpuratus* proteome and contain a N-terminally located CTLL domain coupled to a C-terminal disordered region (Fig 6, Table 2). In contrast *H. erythrogramma* and *H. pulcherrimus* express spicule matrix proteins that feature Type B (PM27) or C (HSM41) conformations (Fig 6) that feature slightly different folded-disordered arrangements. Interestingly, HSM41, which possesses the Type C configuration, evolved with two conserved non-CTLL protein-protein interaction domains instead of the interactive CTLL domain (Fig 6). Again, given the presence of intrinsic disorder, amyloid-like, and protein-protein recognition domains (Fig 4), we hypothesize that the Type B and C spicule matrix proteins would be expected to function as hydrogelators, participate in water exchange, and modulate the mineralization process as per the Type A group. However, the differences in CTLL location and sequence variation as well as the use of non-CTLL interactive domains (Fig 6, Table 2) may reflect molecular strategies that evolved to allow adaptations in hydrogelation, water exchange, and protein-mediated mineral particle assembly that meet the specific skeletal requirements of that organism. Furthermore, as noted for mollusk shells [49], different sea urchin species thrive under different conditions (e.g., water temperature, pH, pressure, salinity, Ca(II)/Mg(II) concentrations and so on) and thus each species may have evolved different primary sequences to perform similar roles in spiculogenesis but "tuned" to the environment of that species. These are intriguing concepts and further research will be required to understand how different sea urchin spicule matrices undergo assembly and mineral formation and how individual proteins participate in this assembly process.

## Acknowledgments

This report represents contribution number 97 from the Laboratory for Chemical Physics, New York University.

## Author Contributions

**Conceptualization:** Martin Pendola, John Spencer Evans.

**Formal analysis:** Martin Pendola, John Spencer Evans.

**Funding acquisition:** John Spencer Evans.

**Investigation:** Martin Pendola, Gaurav Jain, John Spencer Evans.

**Supervision:** John Spencer Evans.

**Validation:** Gaurav Jain.

**Writing – original draft:** John Spencer Evans.

**Writing – review & editing:** John Spencer Evans.

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
