## [Decision Letter · Decision Letter 0]

30 Jul 2019

PONE-D-19-17917

Skeletal development in the sea urchin relies upon protein families that contain intrinsic disorder, aggregation-prone, and conserved globular interactive domains.

PLOS ONE

Dear Dr. Evans,

Thank you for submitting your manuscript to PLOS ONE. After careful consideration, we feel that it has merit but does not fully meet PLOS ONE’s publication criteria as it currently stands. Therefore, we invite you to submit a revised version of the manuscript that addresses the points raised during the review process.

ACADEMIC EDITOR: Please try to improve your manuscript following the advices of one of the reviewers.

We would appreciate receiving your revised manuscript by Sep 13 2019 11:59PM. To enhance the reproducibility of your results, we recommend that if applicable you deposit your laboratory protocols in protocols.io, where a protocol can be assigned its own identifier (DOI) such that it can be cited independently in the future. For instructions see: http://journals.plos.org/plosone/s/submission-guidelines#loc-laboratory-protocols

We look forward to receiving your revised manuscript.

Kind regards,

Eugene A. Permyakov, Ph.D., Dr.Sci.

Academic Editor

PLOS ONE

Journal Requirements:

1. Thank you for including your funding statement; "The funders had no role in study design, data collection and analysis, decision to publish, or preparation of the manuscript."

Please provide an amended Funding Statement that declares *all* the funding or sources of support received during this specific study (whether external or internal to your organization) as detailed online in our guide for authors at http://journals.plos.org/plosone/s/submit-now.  

Please state what role the funders took in the study.  If any authors received a salary from any of your funders, please state which authors and which funder. If the funders had no role, please state: "The funders had no role in study design, data collection and analysis, decision to publish, or preparation of the manuscript."

Reviewers' comments:

Reviewer's Responses to Questions

**Comments to the Author**

1. Is the manuscript technically sound, and do the data support the conclusions?

Reviewer #1: Yes

Reviewer #2: Yes

2. Has the statistical analysis been performed appropriately and rigorously? 

Reviewer #1: Yes

Reviewer #2: Yes

3. Have the authors made all data underlying the findings in their manuscript fully available?

Reviewer #1: Yes

Reviewer #2: Yes

4. Is the manuscript presented in an intelligible fashion and written in standard English?

Reviewer #1: Yes

Reviewer #2: Yes

5. Review Comments to the Author

Reviewer #1: This is an interesting and important study with great potential. The manuscript is well-written and concise. It will have a noticeable impact. However, in my view, it requires additional experiments (a couple of wet lab experiments and one computational experiment).

1) In addition to far-UV CD and 1H-NMR analyses, structure of SpSM50 and SpSM30B/C (S. purpuratus) should be probed by near-UV CD spectroscopy (320-250 nm). This will provide important information on the environment of aromatic residues in the gel state of these proteins.

2) I would recommend to analyze secondary and tertiary structures of SpSM50 and SpSM30B/C (S. purpuratus) in their non-gel states (i.e., under the conditions that do not promote hydrogel particle formation). This can be done using far-UV and near-UV CD spectroscopy, respectively. This analysis will provide an important information on the effect of gelation on structural properties of these proteins.

3) The authors are encouraged to utilized CH-CDF plot method (PMID: 22174269) in analysis of nine spicule matrix proteins listed in Table 1. This will provide an important information on the sub-classification of the overall disorder status of these proteins.

Reviewer #2: This is a nicely constructed and accomplished manuscript. I recommend acceptance of this manuscript for the virtue of its contribution.

6. PLOS authors have the option to publish the peer review history of their article (what does this mean?). If published, this will include your full peer review and any attached files.

Reviewer #1: Yes: Vladimir N. Uversky

Reviewer #2: No

---

## [Author Response · Author response to Decision Letter 0]

21 Aug 2019

Response to reviewers

Reviewer 2

1 The reviewer states, “In addition to far-UV CD and 1H-NMR analyses, structure of SpSM50 and SpSM30B/C (S. purpuratus) should be probed by near-UV CD spectroscopy (320-250 nm). This will provide important information on the environment of aromatic residues in the gel state of these proteins.”

Response: This is a good idea, and in fact in previous papers we have addressed aromatic residues (see Biochemistry 2014, 53, 2739-2748; Biomacromolecules 2014, 15, 4467-4479; ACS Omega 2017, 2, 6151-6158). However, as stated in the introduction (see pp 3,4) our present focus is to address the structural features that exist amongst SM proteins, and, elucidate the basis for spicule matrix protein hydrogel-water binding and release. Moreover, aromatic CD experiments would be qualitative in nature, whereas the NMR experiments that we can employ are a bit more quantitative and thus the Asn, Gln, Arg study is more feasible at the moment. We do intend to study aromatic residues using NMR at a later time.

2 The reviewer states, “I would recommend to analyze secondary and tertiary structures of SpSM50 and SpSM30B/C (S. purpuratus) in their non-gel states (i.e., under the conditions that do not promote hydrogel particle formation). This can be done using far-UV and near-UV CD spectroscopy, respectively. This analysis will provide an important information on the effect of gelation on structural properties of these proteins.”

Response: This is a good idea however, these proteins are very strong aggregators in all buffers and so far we have not been able to identify conditions where a monomeric form of SpSM50 or SpSM30B/C exists, so unfortunately we cannot perform these studies at this time.

3 The reviewer states, “The authors are encouraged to utilized CH-CDF plot method (PMID: 22174269) in analysis of nine spicule matrix proteins listed in Table 1. This will provide an important information on the sub-classification of the overall disorder status of these proteins.”

Response: This is an excellent suggestion and we have now generated a CH-CDF plot (new Figure 5) and discussed its relevance for the spicule matrix proteome (see materials and methods, pp 6-7; results with Figure 5, pg 11; and brief discussion on pg 18).

---

## [Editor Report · Decision Letter 1]

22 Aug 2019

Skeletal development in the sea urchin relies upon protein families that contain intrinsic disorder, aggregation-prone, and conserved globular interactive domains.

PONE-D-19-17917R1

Dear Dr. Evans,

We are pleased to inform you that your manuscript has been judged scientifically suitable for publication and will be formally accepted for publication once it complies with all outstanding technical requirements.

With kind regards,

Eugene A. Permyakov, Ph.D., Dr.Sci.

Academic Editor

PLOS ONE
---

## [Editor Report · Acceptance letter]

23 Sep 2019

PONE-D-19-17917R1 

Skeletal development in the sea urchin relies upon protein families that contain intrinsic disorder, aggregation-prone, and conserved globular interactive domains. 

Dear Dr. Evans:

I am pleased to inform you that your manuscript has been deemed suitable for publication in PLOS ONE. Congratulations! Your manuscript is now with our production department. 

With kind regards,

on behalf of

Prof. Eugene A. Permyakov 

Academic Editor

PLOS ONE